# The Impact of the Intuitionistic Fuzzy Entropy-Based Weights on the Results of Subjective Quality of Life Measurement Using Intuitionistic Fuzzy Synthetic Measure

**DOI:** 10.3390/e25070961

**Published:** 2023-06-21

**Authors:** Ewa Roszkowska, Marzena Filipowicz-Chomko, Marta Kusterka-Jefmańska, Bartłomiej Jefmański

**Affiliations:** 1Faculty of Computer Science, Bialystok University of Technology, Wiejska 45A, 15-351 Bialystok, Poland; e.roszkowska@pb.edu.pl; 2Department of Quality and Environmental Management, Wroclaw University of Economics and Business, 53-345 Wrocław, Poland; marta.kusterka-jefmanska@ue.wroc.pl; 3Department of Econometrics and Computer Science, Wroclaw University of Economics and Business, 53-345 Wrocław, Poland; bartlomiej.jefmanski@ue.wroc.pl

**Keywords:** intuitionistic fuzzy multi-criteria methods, intuitionistic fuzzy entropy-based weights, Intuitionistic Fuzzy Synthetic Measure, subjective quality of life, European cities

## Abstract

In this paper, an extended Intuitionistic Fuzzy Synthetic Measure (IFSM) with intuitionistic fuzzy (IF) entropy-based weights is presented. This method can be implemented in a ranking problem where the assessments of the criteria are expressed in the form of intuitionistic fuzzy values and the information about the importance criteria is unknown. One example of such a problem is measuring the subjective quality of life in cities. We join the debate on the determination of weights for the analysis of the quality of life problem using multi-criteria methods. To handle this problem, four different IF entropy-based weight methods were applied. Their performances were compared and analyzed based on the questionnaires from the survey concerning the quality of life in European cities. The studies show very similar weighting systems obtained by different IF entropy-based approaches, resulting in almost the same city rankings acquired through IFSM by using those weights. The differences in rankings obtained through the IFSM measure (and only by one position) concern the six cities included in the analysis. Our results support the assumption of the equal importance of the criteria in measuring this complex phenomenon.

## 1. Introduction

Global environmental challenges—especially those related to climate change—and social and economic problems of the modern world force decision makers, organizations, communities, and individuals to take action toward sustainable development. These activities can be individual (e.g., changing habits or consumption patterns) or collective, integrating the efforts of businesses, non-governmental organizations, or EU and/or UN governments. The Europe 2020 Strategy is the EU’s agenda adopted for growth and jobs that emphasizes mutually reinforcing the three priorities of smart, sustainable, and inclusive growth [1]. Further, the 2030 Agenda is an extremely important document on the transformation of the modern world toward sustainable development [2]. It is a global development strategy that lasts until 2030 and consists of 17 sustainable development goals and 169 tasks. It has been adopted by 193 UN member states.

In this paper, we concentrate on sustainable development in the area of subjective quality of life in cities. According to Agenda 2030, goal 11 assumes that by 2030 cities and human settlements should be inclusive, safe, resilient, and sustainable. Cities must therefore meet the urbanization challenges related to the management and optimal involvement of local resources and the social, environmental, and economic potential of a given territorial unit. Paraphrasing the definition of sustainable development from the Brundtland Report [3], it can be assumed that a sustainable city meets the needs of its current residents without compromising the chances of future generations to meet their needs. The notion of quality of life relates to the satisfaction level of meeting human needs and the general level of satisfaction with life and its specific spheres [3].

The measurement of sustainable development is a complex problem. Therefore, several propositions on monitoring the level of sustainability assessment using multi-criteria methods or fuzzy multi-criteria methods can be found in the literature [4,5,6,7,8,9,10,11,12]. Consequently, different multi-criteria decision-making (MCDM) methods were also employed to assess the quality of life. Concerning the quality of life, in some studies, the Analytic Hierarchy Process (AHP) method was used [13,14]. Vakilipour et al. [15] evaluated the quality of life by using the Technique for Order Preferences by Similarity to Ideal Solution (TOPSIS), VlseKriterijumska Optimizacija Kompromisno Resenje (VIKOR), Simple Additive Weighted (SAW), and Elimination and Choice Translating Reality (ELECTRE). In another study, Özdemir Işık and Demir [16] utilized a combination of the AHP for the primary criteria and ELECTRE for the sub-criteria to rank all the indicators. Meanwhile, Kaklauskas et al. [17] included the quality of life as an indicator of a sustainable city, using the Quality of Life Index (QLI) and INVAR techniques along with other indicators. Gonzalez et al. [18] applied the Data Envelopment Analysis (DEA) and Value Efficiency Analysis (VEA) techniques to consolidate information and formulate an index for urban life quality. Roszkowska et al. [19] applied the fuzzy intuitionistic TOPSIS (IF-TOPSIS) method to rank European cities for their quality of life, while in [11] the Intuitionistic Fuzzy Synthetic Measure (IFSM) was utilized.

A crucial stage in the implementation of most MCDM methods for the construction of synthetic measures or composite indicators of such measures is the selection of the importance weights for the different criteria (individual indicators) [10,20,21]. This is because the adopted weights may significantly affect the results of the ranking. In the multi-criteria approach, the weights can be classified as subjective or objective, depending on the method of information acquisition [22,23,24,25,26]. The subjective weights are obtained from subjective preference information on criteria given by the decision maker (DM), while the objective weights are calculated from the information in a decision matrix using mathematical models. In applying the multi-criteria approach for the measurement of complex social phenomena, some authors also discuss the usability of equal weights [27,28]. Maggino and Ruviglioni [28] observed that “equal weighting represents the preferred procedure, adopted in most of the applications”.

Recommendations on the weighting of individual indicators can be found, among others, in OECD’s Handbook on Constructing Composite Indicators [9]. They refer to both formal methods, including e.g., factor analysis, data envelopment analysis, analytic hierarchy processes, conjoint analysis as well as weights based only on expert opinion. Gan et al. [29] described nine frequently used weighting methods and discussed their advantages and disadvantages. Moreover, Greco et al. [10] analyzed e.g., different methodological approaches to weighting the variables that make up the composite index. As the authors of the study emphasize, none of the approaches is free from criticism and each has specific advantages and disadvantages. Therefore, it is not possible to indicate the best way to determine the weights when constructing a composite indicator.

Extensive discussion concerning weights is also provided in analyzing complex problems concerning quality of life. Hagerty et al. [27] reviewed 22 quality of life indicators proposed and used by governments and international institutions. The authors of this study recommended the use of weights in the construction of composite indicators of the quality of life and the use of two-stage factor analysis with conjoint analysis to determine them. Decanacq & Lugo [21] analyzed eight approaches to set weights in the construction of a composite indicator. The authors categorized the approaches into three classes: data-driven, normative, and hybrid weighting. They critically surveyed these approaches and compared their respective advantages and disadvantages. Hagerty & Land [20] provided a framework to jointly consider weights and social indicators in constructing a composite quality of life index that would be approved both by individuals and by researchers. As noted by Hagerty & Land [20] and González et al. [30], it would be ideal to include in the construction of the composite indicator the weights that each respondent assigns to a given quality of life dimension. Individuals are the final consumers and ultimate arbiters of their sense of quality of life,, which is why it is crucial to investigate how individuals themselves weigh various indicators and fields to judge their quality of life [20]. This possibility is provided by OECD Better Life Index. It allows each of the respondents to individually assign importance to particular areas that make up the Better Life Index. This type of approach results from the assumption that satisfying various needs may, to a varying extent, contribute to improving the quality of life of both individuals and entire communities. According to some authors [3], it is important to create such tools for measuring the quality of life that will allow us to take into account the importance that individual respondents attribute to particular aspects of the quality of life.

One of the methods of measuring the quality of life is using data from public statistics sources and surveys of residents’ opinions [31,32,33,34]. Since 2004, the European Commission has provided a survey concerning the quality of life in European cities among the inhabitants of capital cities and large cities located in the European Union, EFTA, Western Balkans, Great Britain, and Turkey. This makes it possible to monitor changes in the subjective quality of life and satisfaction of residents with various aspects of the functioning of cities. Jefmański et al. in [11,35] proposed the Intuitionistic Fuzzy Synthetic Measure (IFSM) for measuring the subjective quality of life based on these surveys. The IFSM aggregated ordinal data using an intuitionistic fuzzy approach handling uncertainty expressed by the respondents’ indecisiveness in assessing selected aspects of the quality of life. In Jefmański et al.’s studies [11,35], equal weights were adopted for the individual dimensions of subjective quality of life. This article is a continuation of research in the field of IFSM application in the measurement of subjective quality of life. In previous studies, the same weights were used for the criteria describing the subjective quality of life. Therefore, the research gap is the assessment of the impact of weighting systems on the stability of IFSM results. We applied a well-known concept of entropy-based weights [22,36,37,38] in the intuitionistic fuzzy approach for evaluating the importance of criteria assuming unknown information about them. Let us observe that the source of a such situation may be time pressure, lack of knowledge or data, and the decision maker’s limited expertise on the problem under consideration.

The main aim of the paper is to propose an extension of the IFSM measure introducing IF entropy-based weights. It should be noticed that different intuitionistic fuzzy (IF) entropy-based methods can be used by researchers and practitioners [39], who applied different ideas for determining the weights. However, there is a lack of research that compares the performance of different IF entropy-based weight methods in real-life problems. To address this problem, in this study, four different entropy-based weight methods were added to the complex problem of quality of life. Their performances were compared and analyzed based on the questionnaires from the survey concerning the quality of life in European cities.

The objectives and contributions of this study are the following:Extension of IFSM measure introducing IF entropy-based weights for the evaluation of socio-economic phenomena with survey data;Comparison of the performance of different IF entropy-based weight methods in the problem of the measurement of the importance of individual indicators of subjective quality of life;Comparison of the performance IFSM by applying different IF entropy-based weight methods in the problem of subjective quality of life measurement.

The rest of this article is organized as follows: In Section 2, the basic concepts related to intuitionistic fuzzy sets (IFS), distances on IFSs, and entropy measures are presented. The general framework of the IFSM with intuitionistic fuzzy entropy-based weights is proposed in Section 3. Section 4 discusses the problem of the application of different IF entropy-based weight techniques in terms of the subjective criteria of quality of life and rank-ordering of the cities using the IFSM measure. Finally, we present the conclusions of the paper.

## 2. Preliminaries

This section presents a brief introduction of basic notions used in the analysis, i.e., intuitionistic fuzzy set (IFS), distances between IFS, and entropy measure.

### 2.1. The Intuitionistic Fuzzy Set

In 1986, Atanassov [40] extended the concept of Zadeh’s [41] fuzzy set (FS) to an intuitionistic fuzzy set (IFS) by introducing a non-membership degree and hesitance degree. The IFS theory is a useful tool to deal with uncertainty and incomplete or imprecise information in the decision-making process more effectively than can be done using FSs [42,43,44].

**Definition** **1**([40,45]). *An Intuitionistic Fuzzy Set (IFS) is defined as:*(1)$$A=\left\{<x,\mu_{A}(x),\nu_{A}(x)>|x\in X\right\}$$
*where*
$\mu_{A}:X\to [0,1]$
*,*
$\nu_{A}:X\to [0,1]$
*and*
(2)$$0\leq \mu_{A}(x)+\nu_{A}(x)\leq 1.$$

The numbers $\mu_{A}(x)$ and $\nu_{A}(x)$ represent the degrees of membership and non-membership of the element x∈X to the set *A*; $\pi_{A}(x)=1-\mu_{A}(x)-\nu_{A}(x)$ the intuitionistic fuzzy index or hesitancy degree of the element x in set A.

If the universe X contains only one element x, then IFS A is denoted by $A=(\mu_{A},\nu_{A})$ and called an intuitionistic fuzzy value (IFV) [46,47].

### 2.2. The Weighted Euclidean and Hamming Distances Measure between IFS

A lot of distance measures between IFSs have been proposed in the literature [48,49,50,51,52,53,54]. Among them, widely used are Hamming and Euclidean distances based on two or three parameters [50]. In this paper, we used the concept of weighted Hamming and Euclidean distances proposed by Xu [53].

**Definition** **2**([53]). *Let us consider*
A,B∈IFS
*with membership functions*
$\mu_{A}(x)$*,*
$\mu_{B}(x)$
*and non-membership functions*
$\nu_{A}(x)$*,*
$\nu_{B}\left(x\right)$*, respectively.**Then, the weighted Euclidean distance between A and B is calculated as follows:*(3)$$dE\left(A,B\right)=\sqrt{\frac{1}{2}\sum_{j=1}^{m}w_{j}[(\mu_{A}(x_{j})-\mu_{B}(x_{j}){)}^{2}+(\nu_{A}(x_{j})-\nu_{B}(x_{j}){)}^{2}+(\pi_{A}(x_{j})-\pi_{B}(x_{j}){)}^{2}]}$$*and the weighted Hamming distance between A and B is defined as follows:*(4)$$dH\left(A,B\right)=\sum_{j=1}^{m}w_{j}[|\mu_{A}\left(x_{j}\right)-\mu_{B}\left(x_{j}\right)|+|\nu_{A}\left(x_{j}\right)-\nu_{B}\left(x_{j}\right)|+|\pi_{A}\left(x_{j}\right)-\pi_{B}\left(x_{j}\right)|],$$*where* $\sum_{j=1}^{m}w_{j}=1$.

The membership degree, non-membership degree, and hesitancy degree together allow for a more precise and complete expression of the decision-making context. The hesitancy degree describes the lack of information in the decision-making process.

### 2.3. The Notion of Intuitionistic Fuzzy Entropy

The entropy concept is used for measuring uncertain information. Shannon [55] proposed a probabilistic entropy function for a measure of uncertainty in a discrete distribution based on the Boltzmann entropy of classical statistical mechanics. The classical entropy approach has been extended by Zadeh [56] to measure fuzziness using probability methods. Subsequently, many entropy measures associated with a fuzzy set or intuitionistic fuzzy set have been proposed in the literature. De Luca and Termini [57], based on Shannon’s probability entropy, proposed an axiomatic framework for the non-probabilistic entropy formula of a fuzzy set, interpreting it as a measure of the amount of information. The intuitionistic fuzzy entropy measure has been investigated by Burillo and Bustince [58], Szmidt and Kacprzyk [59], Hung and Yang [60], Vlachos and Sergiadis [44], Ye [61], Guo [62], Yuan and Zheng [63], Liu [64], Khaleie and Fasanghari [65], among others.

Burillo and Bustince [58] proposed the notion of entropy measuring the degree of hesitancy of IFS. Szmidt and Kacprzyk [59] defined a non-probabilistic entropy measure with a geometric interpretation of IFSs and used a ratio of distances between them. Hung and Yang [60] extended the De Luca and Termini [57] axiom definition to IFSs and proposed two families of entropy measures on IFSs. Vlachos and Sergiadis [44] introduced an approach discrimination measure for IFS based on the information. The intuitionistic fuzzy entropy measure was composed of the hesitancy degree and the fuzziness degree of the IFS. Moreover, cross-entropy in the intuitionistic fuzzy setting as an extension of the De Luca–Termini [57] non-probabilistic entropy for IFSs was proposed. Chen and Li [39] adopted Vlachos and Sergiadis’ [66] proposition of the interval intuitionistic fuzzy entropy measure in an intuitionistic environment and implemented the axiomatic approach. Ye [61] investigated two measures of intuitionistic fuzzy entropy that are a generalized version of the fuzzy entropy investigated by Parkash et al. [67].

We listed here four intuitionistic entropy measures that will be used later to determine the weights of the criteria [39,44,58,59]. These methods were selected because they have been used widely in applications. Moreover, they represent different approaches for measuring entropy in an intuitionistic fuzzy context.

**Definition** **3.***Let* $A=\left\{<x,\mu_{A}(x),\nu_{A}(x)>|x\in X\right\}$ *be the set of IFS defined on the universe of discourse* $X=\left\{x_{1},x_{2},\ldots ,x_{n}\right\}.$ *Then, the entropy measure* $E_{i}\left(A\right)$ *(for* i=1,2,3,4*) is defined as follows [58,59]:*(5)$$E_{1}\left(A\right)=\sum_{i=1}^{n}(1-(\mu_{A}{(x}_{i})+\nu_{A}{(x}_{i}))=\sum_{i=1}^{n}\pi_{A}{(x}_{i})$$(6)$$E_{2}(A)=\frac{a}{b}$$ *where* a *is the distance (A;* $A_{near}$*) from A to the nearer point* $A_{near}$ *among points *(1,0,0)* and *(0,1,0)*, and* b *is the distance (A;* $A_{far}$*) from A to the farther point* $A_{far}$ *among points *(1,0,0)* and *(0,1,0)*;*(7)$$E_{3}\left(A\right)=\frac{\overset{n}{\underset{i=1}{\sum }}\left(min(\mu_{A}{(x}_{i}),\nu_{A}{(x}_{i}))+min(1-\mu_{A}{(x}_{i}),1-\nu_{A}{(x}_{i}))\right)}{\overset{n}{\underset{i=1}{\sum }}\left(max(\mu_{A}{(x}_{i}),\nu_{A}{(x}_{i}))+max(1-\mu_{A}{(x}_{i}),1-\nu_{A}{(x}_{i}))\right)}$$(8)$$E_{4}(A)=-\frac{1}{nln2}\sum_{i=1}^{n}[\mu_{A}\ln\mu_{A}+\nu_{A}ln\nu_{A}-\left(1-\pi_{A}\right)ln(1-\pi_{A})-\pi_{A}ln2]$$*where:* $\frac{1}{nln2}-$ *constant which assures* $0\leq E_{4}(A)\leq 1,j=1,2,\ldots ,n.$

Let us notice that using Euclidean distance, we have [59]:(9)$$E_{2E}\left(A\right)=\frac{\sqrt{\overset{n}{\underset{i=1}{\sum }}\left[{\left(\mu_{near}{(x}_{i})-\mu_{A}{(x}_{i})\right)}^{2}+{\left(\nu_{near}{(x}_{i})-\nu_{A}{(x}_{i})\right)}^{2}+{\left(\pi_{near}{(x}_{i})-\pi_{A}{(x}_{i})\right)}^{2}\right]}}{\sqrt{\overset{n}{\underset{i=1}{\sum }}\left[{\left(\mu_{far}{(x}_{i})-\mu_{A}{(x}_{i})\right)}^{2}+{\left(\nu_{far}{(x}_{i})-\nu_{A}{(x}_{i})\right)}^{2}+{\left(\pi_{far}{(x}_{i})-\pi_{A}{(x}_{i})\right)}^{2}\right]}},$$
and using Hamming distance, we have [39,44,59]:(10)$$E_{2H}\left(A\right)=\frac{\overset{n}{\underset{i=1}{\sum }}\left|\mu_{near}{(x}_{i})-\mu_{A}{(x}_{i})\right|+\left|\nu_{near}{(x}_{i})-\nu_{A}{(x}_{i})\right|+\left|\pi_{near}{(x}_{i})-\pi_{A}{(x}_{i})\right|}{\overset{n}{\underset{i=1}{\sum }}\left|\mu_{far}{(x}_{i})-\mu_{A}{(x}_{i})\right|+\left|\nu_{far}{(x}_{i})-\nu_{A}{(x}_{i})\right|+\left|\pi_{far}{(x}_{i})-\pi_{A}{(x}_{i})\right|}$$

It is also worth noticing that we can rewrite Formula (9) as:(11)$$E_{4}(A)=E_{fuzz}(A)+E_{int}(A)$$
where
(12)$$E_{fuzz}(A)=-\frac{1}{nln2}\sum_{i=1}^{n}[\mu_{A}ln\mu_{A}+\nu_{A}ln\nu_{A}-\left(1-\pi_{A}\right)ln(1-\pi_{A})]$$
and
(13)$$E_{int}(A)=\frac{\overset{n}{\underset{i=1}{\sum }}\pi_{A}{(x}_{i})}{n}$$

Let us observe that $E_{int}(A)$ is the normalized entropy proposed by Burillo and Bustince [58]. Therefore, intuitionistic fuzzy entropy consists of two intuitively and mathematically distinct components: one expressing the degree of fuzziness and the other the degree of intuitionism of an IFS *A* [44]. In FSs theory, entropy is indeed a measure of fuzziness, while IFSs’ entropy $E_{4}(A)$ is a measure of both fuzziness and intuitionism.

### 2.4. The Objective Entropy Weight under an Intuitionistic Fuzzy Environment

The proper assessment of criteria weights is an important part of the algorithm of MCDM methods for analyzing complex problems. The systems of weights may significantly influence the final rankings of alternatives. As was mentioned in the Introduction, the weights can be classified as subjective or objective, depending on the method of information acquisition. When the information about criteria weights are completely unknown or incompletely known, the entropy-based techniques, with their modifications and extensions, are widely used [36]. It should be noticed that IF entropy-based weighting methods depend on the nature of a decision matrix in an intuitionistic fuzzy environment [39]. They focus on the discrimination among data or the credibility of data to determine criteria weights. The advantage of the entropy-based methods is that the calculation of the weight is simple and uncomplicated and uses only information provided by the criteria. Let us note that the smaller entropy value for criterion across alternatives provided the decision maker with the most useful information according to the entropy theory. Therefore, the criterion should be assigned a larger weight. Conversely, the larger the entropy value for criterion across alternatives provided to the decision maker, the less useful the information. In such a situation, a criterion should be evaluated with a smaller weight [39,63,68].

Chen and Li [39] classified IF entropy measures as useful for calculating the criteria weights concerning the hesitancy degree, probability, non-probability, and geometry. A comparative analysis of different measures to generate weights was also provided by the computational experiment. The results of the experiment and Pearson correlations, Spearman rank correlations, contradiction rates, inversion rates, and consistency rates show that ranking the alternatives depends on the type of IF entropy measures as well as the number of criteria and alternatives.

Ye [69] proposed a multi-criteria fuzzy decision-making method based on the weighted correlation coefficient using entropy weights under an intuitionistic fuzzy environment. Burillo and Bustince’s [58] entropy formula has been used for determining the criteria weights with completely unknown information about criteria weights in cases where values take the form of intuitionistic fuzzy numbers. Hung and Chen [68] proposed a fuzzy TOPSIS decision-making model using Vlachos and Sergiadis’s [44] concept of entropy for dealing with multiple criteria decision-making (MCDM) problems under an intuitionistic fuzzy environment. This model allows for measuring the degree of satisfiability and the degree of non-satisfiability, respectively, of each alternative evaluated across a set of criteria.

Jun Ye [70] applied entropy weight models to determine the weights from intuitionistic fuzzy decision matrices. Kumar et al. [38] conducted a literature review concerning the application of entropy weight methods for multi-objective optimization in machining operations. Zhang et al. [71] applied an intuitionistic fuzzy entropy weight method in a service supply chain optimization.

The number of propositions that the intuitionistic entropy can measure has substantially increased in the last twenty years. This observation motivated the authors to make a comparison of the most representative entropy-based weight methods in practical application. Although the entropy-based weights were applied to solve real problems, this work extends the analysis in the context of the analysis of quality of life and examines the assumptions concerning equal weights in the IFSM measure applied to analyze the survey [11,35].

## 3. The Intuitionistic Fuzzy Synthetic Measure with IF Entropy-Based Weights

The Intuitionistic Fuzzy Synthetic Measure (IFSM) has been proposed by Jefmański et al. [35] for analyzed survey responses represented by ordinal data. The IFSM has been inspired by Hellwig’s measure [72] and adapted to an intuitionistic fuzzy environment. This method used the idea of the measure distances from the object to the pattern development (pattern object, reference point). The other most popular multi-criteria methods that implemented reference points are TOPSIS [36,73] and VIKOR [74,75]. The main use of TOPSIS is to select the alternative (object) that is closest to the ideal solution (pattern object) and furthest from the negative ideal solution (anti-pattern object). The VIKOR procedure uses the multiple criteria ranking index based on the measure of “closeness” to the “ideal” solution (pattern object). Each alternative (object) is evaluated concerning each criterion, and then the compromise ranking can be obtained by comparing the relative closeness measure to the ideal alternative (pattern object).

In contrast, Hellwig’s method uses only the pattern object concept. Hellwig’s classic algorithm was presented by UNESCO [76] (1972a) as a tool for measuring health development in a group of countries. Over the years, this procedure [72] has been applied in many areas [77,78,79,80,81] and modified for real data [82], fuzzy sets [83,84], intuitionistic fuzzy sets [11,35], interval-valued fuzzy sets [85], and ordered fuzzy sets [86]. It is worth noting that TOPSIS and VIKOR methods were also adapted to the intuitionistic environment (see e.g., [19,87,88] and [89,90], respectively).

In this paper, we extended the IFSM [35] method by introducing entropy-based weights in the algorithm. Assuming that the information about the importance of criteria is completely unknown, we applied a concept of intuitionistic fuzzy entropy (see Section 2.3) for determining the criteria weights. We adopt the weighting procedure proposed by Chen and Li [39].

Let $O=\left\{0_{1},0_{2},\ldots ,0_{n}\right\}(i=1,2,\ldots ,n)$ be the set of objects under evaluation, and let $C=\left\{C_{1},C_{2},\ldots ,C_{m}\right\}(j=1,2,\ldots ,m)$ be the set of criteria. The characteristic of the *i*-th object in terms of the *j*-th criterion is expressed in the form of an intuitionistic fuzzy value $\left(\mu_{ij},\nu_{ij}\right)$, and $\pi_{ij}=1-\mu_{ij}-\nu_{ij}$. Therefore, the *i*-th object is represented by the vector:(14)$$O_{i}=[(\mu_{i1},\nu_{i1}),\ldots ,(\mu_{im},\nu_{im})],$$
where i=1,2,…,n.

The IFSM procedure with IF entropy-based weights consists of the following steps:**Step 1.** Determination of the Intuitionistic Fuzzy Decision Matrix.
(15)$$D=\left[\begin{array}{cccc} \left(\mu_{11},v_{11}\right) & \left(\mu_{12},v_{12}\right) & \cdots & \left(\mu_{1n},v_{1m}\right)\\ \left(\mu_{21},v_{21}\right) & \left(\mu_{22},v_{22}\right) & \cdots & \left(\mu_{2n},v_{2m}\right)\\ \vdots & \vdots & \ddots & \vdots \\ \left(\mu_{n1},v_{n1}\right) & \left(\mu_{n2},v_{n2}\right) & \cdots & \left(\mu_{nm},v_{nm}\right) \end{array}\right]$$
where $\left(\mu_{ij},\nu_{ij}\right)$—intuitionistic fuzzy evaluation of the *i*-th object in terms of the *j*-th criterion (i=1,2,…,n;j=1,2,…,m). The method of determining intuitionistic fuzzy values and building the matrix is described in detail in the study [35].**Step 2.** Determination of the vector of weights.The stage of determining IF entropy-based weights [39] consists of the following steps:**Step 2a:** Transformation of the matrix *D* (see Formula (15)) to IF entropy matrix *ED*:
(16)$$ED=\left[\begin{array}{cccc} E_{11} & E_{12} & \cdots & E_{1m}\\ E_{21} & E_{22} & \cdots & E_{2m}\\ \vdots & \vdots & \ddots & \vdots \\ E_{n1} & E_{n2} & \cdots & E_{nm} \end{array}\right]$$
where $E_{ij}$ is an entropy value of IF value $\left(\mu_{ij},v_{ij}\right)$ from the matrix *D* calculated using one of the Formulas (5)–(10), (i=1,2,…,n; j=1,2,…,m).**Step 2b**: Aggregation and normalization of the IF entropy values using the formula:
(17)$$S_{j}=\frac{\overset{n}{\underset{i=1}{\sum }}E_{ij}}{\overset{m}{\underset{j=1}{\sum }}\overset{n}{\underset{i=1}{\sum }}E_{ij}}$$ where i=1,2,…,n; j=1,2,…,m.**Step 2c:** Calculation of the criteria weights using the formula:
(18)$$w_{j}=\frac{1-S_{j}}{\overset{m}{\underset{j=1}{\sum }}\left(1-S_{j}\right)}$$ where j=1,2,…,m.Finally, the weighs vector has the form $w=[w_{1},\ldots ,w_{m}]$, where
(19)$$w_{j}\in [0,1]\;\mathrm{and}\;\sum_{j=1}^{m}w_{j}=1.$$**Step 3.** Determination of the intuitionistic fuzzy pattern object $\left(I_{IFI}\right)$ as:
(20)$$I_{IFI}^{}=[(\underset{i}{max}\mu_{i1},\underset{i}{min}\nu_{i1}),\ldots ,(\underset{i}{max}\mu_{im},\underset{i}{min}\nu_{im})]$$ where $\left(\mu_{ij},\nu_{ij}\right)$, denote the evaluation information of the *i*-th object with respect to the *j*-th criterion and $\pi_{ij}=1-\mu_{ij}(x)-\nu_{ij}(x)$ (i=1,2,…,n; j=1,2,…,m).**Step 4** Calculation of the distance measures between the objects and the intuitionistic fuzzy pattern object using either Formula (3) or (4).The distance measure from the pattern object takes the form of:(21)$$d^{+}(O_{i})=d(I_{IFS},O_{i})$$
where i=1,2,…,n.**Step 5.** Calculation of the value of the Intuitionistic Fuzzy Synthetic Measure.The Intuitionistic Fuzzy Synthetic Measure (*IFSM*) coefficient is defined as follows:(22)$$IFSM(O_{i})=1-\frac{d^{+}(O_{i})}{d_{0}}$$
where: $d_{0}={\bar{d}}_{0}+2S(d_{0})$, ${\bar{d}}_{0}=\frac{1}{n}\sum_{i=1}^{n}d^{+}(O_{i})$, $S(d_{0})=\sqrt{\frac{1}{n}\sum_{i=1}^{n}\left(d^{+}(O_{i}\right)-{\bar{d}}_{0}{)}^{2}}$.**Step 6.** Rank ordering of objects by maximizing the coefficient $IFSM(O_{i})$.The highest value of $IFSM(O_{i})$, then the highest position of the object $O_{i}(i=1,2,\ldots ,n)$. The procedure steps for IFSM with IF entropy-based weights are presented in Figure 1.

## 4. Application of IFSM with IF Entropy-Based Weights in Assessing the Quality of Life in Selected European Cities

### 4.1. Problem Description

The European Commission authorized IPSOS to conduct surveys on the quality of life in European cities [91]. Until now, five editions of subjective quality of life surveys of residents of selected European cities have been carried out. Responsibility for local development and the quality of life of residents rests with local authorities, which should undertake and implement actions that will allow the optimal use of the city’s development potential. As important centers of business, non-governmental organizations, and public institutions, cities have great potential for the development of innovation and new technologies. On the other hand, cities are primarily places to live, offering their residents many amenities, such as access to the labor market, education, health care, sports infrastructure, and leisure activities. Living in the city, however, comes with some nuisances. These result from, for example, the high congestion in cities and the degradation of the urban environment due to air, water, and soil pollution, as well as noise emissions and waste. Different types of social problems also exist in cities, resulting from inequalities in the level of wealth of residents and their access to services, as well as the consequences resulting from these realities—social exclusion, lack of a sense of local identity, a low sense of security, and low social trust.

To meet all the challenges arising from the urbanization processes and the accompanying social, economic, and environmental phenomena, cities should constantly monitor their development using both objective indicators and research on the subjective quality of life of residents. The periodically conducted surveys of satisfaction perceived by city dwellers with particular areas of the functioning of the urban fabric are a barometer of the mood of the local community and a signpost for the city authorities by which the direction of a given unit should develop. In addition, these surveys allow us to identify the best-rated areas of city functioning and those that require improvement. Information from this type of research can be used by local authorities to shape and implement local development policy. Finally, conducting comparative research on cities yields important information for both current and potential residents, entrepreneurs, and tourists.

### 4.2. Data Source

The fifth edition survey on quality of life in European cities covered 83 cities across the EU, EFTA countries, the UK, Western Balkans, and Turkey [91]. The study was carried out from 12 June to 27 September 2019, with a break from 15 July to 1 September. A total of 700 interviews were conducted in each of the surveyed cities, resulting in a total of 58,100 participants (see also [11,35]).

The study aimed to measure and compare inhabitants’ satisfaction with life in selected cities using the IFSM method with different IF entropy-based weights. Inhabitants’ satisfaction, as a complex phenomenon, was characterized using 10 criteria:

C_1_—satisfaction with public transport.

C_2_—satisfaction with health care services, doctors, and hospitals.

C_3_—satisfaction with sports facilities such as sports fields and indoor sports halls.

C_4_—cultural facilities such as concert halls, theaters, museums, and libraries.

C_5_—satisfaction with green spaces such as parks and gardens.

C_6_—satisfaction with public spaces such as markets, squares, pedestrian areas.

C_7_—satisfaction with schools and other educational facilities.

C_8_—satisfaction with the quality of the air.

C_9_—satisfaction with the noise level.

C_10_—satisfaction with the cleanliness.

In assessing the criteria, a 5-point measurement scale was used: very satisfied, rather satisfied, rather unsatisfied, very unsatisfied, don’t know/no answer.

An example of the assessment of London in terms of 10 criteria using the 5-point Likert scale is presented in Table 1.

### 4.3. Results and Discussion

First, the respondents’ assessments were transformed into IFVs. For each criterion, the parameter ν has been calculated as the sum of categories 1 and 2 in the survey, μ as the sum of categories 3 and 4, and π as category 99. Subsequently, the obtained data were divided by 100 to obtain a value ranging from 0 to 1. The transformations of the respondents’ assessments of London (Table 1) into IFVs are presented in Table 2. For instance, for criterion 1 the degree of non-membership is calculated as (2.475 + 11.307)/100 = 0.13782, the degree of membership as (41.520 + 42.453)/100 = 0.83953, and the degree of hesitancy as 2.244/100 = 0.0224 (see Table 1 and Table 2).

Criteria assessments in the form of IFVs for all the analyzed cities are presented in Table A1, Table A2 and Table A3 (see Appendix A).

In step 2, we assign weights for individual criteria according to Formulas (5) and (7)–(9), applying four IF entropy-based weight techniques. The results are presented in Table 3. Because we later calculate the value of IFSM using Euclidean distance, we also applied the weight *E*2 entropy measure defined by Formula (9), based on the Euclidean entropy measure. Four IF entropy-based systems of weights (*E*1–*E*4) were compared with equal weights (*E*0).

From Table 3, we can conclude that despite the use of different formulas in the calculation of entropy measures, the values of the weight coefficients of the criteria are at a very similar level. The differentiation of the weight coefficients between the criteria is very low.

To compare the systems of weights, Garuti’s G compatibility index [92,93] was utilized, which is defined as follows:(23)$$G_{Es}^{Er}=\overset{n}{\underset{j=1}{\sum }}\left[\frac{min(w_{j}^{Er},w_{j}^{Es})}{max(w_{j}^{Er},w_{j}^{Es})}\cdot \frac{\left(w_{j}^{Er}+w_{j}^{Es}\right)}{2}\right]$$
where r,s=0,1,2,3,4.

The index $G_{Es}^{Er}$ = 1 indicates the total compatibility of two systems of weights, while $G_{Es}^{Er}$ = 0—total incompatibility.

The results of applying Garuti’s compatibility index for systems of weights are presented in Table 4. We can confirm high compatibility between any two systems of weights.

In step 3, the intuitionistic fuzzy pattern object has been determined (see Table 5).

In step 4, the values of distance measures between the objects and the intuitionistic fuzzy pattern object $I_{IFI}$ are calculated. In step 5, the values of IFSM are calculated. Table 6 summarizes the values of *IFSM* and the rankings of cities obtained by using systems of weights, as presented in Table 3. For simplicity, we refer to IFSM*i* for the coefficient obtained using a system of weights *Ei* (i=0,1,2,3,4).

While analyzing the positions of cities in the overall rankings obtained using different synthetic measures, one may observe that the rankings of six cities change (highlighted in grey in Table 6), but by a maximum of one position, which is confirmed by the Spearman coefficient (Table 7). What is interesting is that the differences between the rankings are observed only for the *IFSM*1 measure when compared with others. Let us note that the weights system in the IFSM1 method is based on Formula (5), which takes into consideration only hesitancy. Our results support the assumption of the equal importance of the criteria while measuring the quality of life based on the survey [11,35].

Further, the differences in IFSM values are very small, which is confirmed by Pearson’s coefficients (Table 8). In both cases, the differences in coefficients can be observed on the 4th, 5th, or 6th decimal place.

Basic descriptive statistics for all Intuitionist Fuzzy Synthetic Measures are presented in Table 6 and Figure 2.

When analyzing the box plots of the *IFSM* values, it is noticeable that the *IFSM* values are extremely similar. The differences in the *IFSM* values of the cities are about 0.9. The mean value is about 0.5, with a SD of about 0.25.

In summary, the application of the criteria weighting method based on different formulas for the IF entropy measure did not significantly affect the IFSM measure values and the cities’ positions in the rankings. In each of the rankings obtained, the cities with the highest levels of subjective quality of life were, in order, Luxembourg, Wienna and Helsinki. The values of the IFSM measure of each of these cities exceeded the 0.8. The lowest-ranked cities were, in order, Palermo, Athina, and Skopje. The values of the *IFSM* measures of cities were very diverse and were within the entire allowable range. An interesting case of the city of Palermo was also observed, for which some of the values of the *IFSM* measure were negative. This is a rare but possible case when the value of the IFSM measure may go beyond the [0;1] range. In such a situation, the Double Intuitionistic Fuzzy Synthetic Measure [44] can be calculated, which, by using an additional anti-pattern object in its formula, normalizes the results in the range [0;1]. As previously noted, the use of different measures of entropy had little effect on cities’ rankings. Only in the cases of six cities (Ankara, Ljubljana, Paris, Prague, Stockholm, and Zagreb), were changes in the position of the rankings observed (always by only one position). No particular regularities were noticed in this regard, as changes in the rankings concerned both relatively high-ranking and medium-ranking cities. In the cases of cities with the lowest levels of subjective quality of life, no changes in rankings were observed.

## 5. Conclusions and Future Research

In this paper, three main scientific goals have been achieved. The first goal was to extend IFSM for IF entropy-based weights, which would allow handles to lack knowledge about importance criteria when they are evaluated using intuitionistic fuzzy values. Let us emphasize that the major advantage of entropy-based weight methods is simple and uncomplicated calculations using only information provided by the criteria and an intuitive interpretation of the entropy measure.

The second goal of the paper was to verify the proposed framework by implementing it to measure the subjective quality of life. The IFSM method was used in the analysis of the results from the fifth survey on quality of life in selected European cities.

The final goal was to compare the performance of different IF entropy-based weight methods in the problem of measuring the importance of individual criteria of subjective quality of life. We also compared the rankings obtained by means of IFSM, with different IF entropy-based weight methods.

An analysis conducted on quality of life measurement results confirmed and illustrated the effectiveness of the proposed method. The application of the weighting method based on entropy measures did not have a major impact on the results of measuring the subjective quality of life in selected European capitals using the IFSM. This confirms the validity of the approach adopted by the authors in previous works [35] regarding the measurement of the subjective quality of life in European cities, where the validity of the criteria was not differentiated. This approach simplifies the measurement itself and the analysis of the results obtained, while providing stable and reliable measurement results.

A certain weakness of the article is the lack of comparisons of the adopted weighting method based on entropy measures with other weighting methods. Therefore, future research will focus on several areas: The first will concern the assessment of the impact of other objective weighting methods on the results of measuring the subjective quality of life using IFSM. Carrying out comparative studies is also being planned; these studies will compare the IFSM method with other methods based on reference points, such as intuitionistic fuzzy VIKOR or intuitionistic fuzzy TOPSIS, in the context of their sensitivity to the choice of weighting method. Furthermore, the theoretical framework of the IFSM with different IF entropy-based weights needs to be investigated. In addition, it should be emphasized that the subject of the article was only objective weights. Comparing them with the subjective weights obtained on the basis of the opinions of European city dwellers could provide additional guidance on the necessity and ways of determining the importance of the criteria. Finally, the proposed method should be utilized in other applications for further verification of its effectiveness.

## Figures and Tables

**Figure 1 entropy-25-00961-f001:**
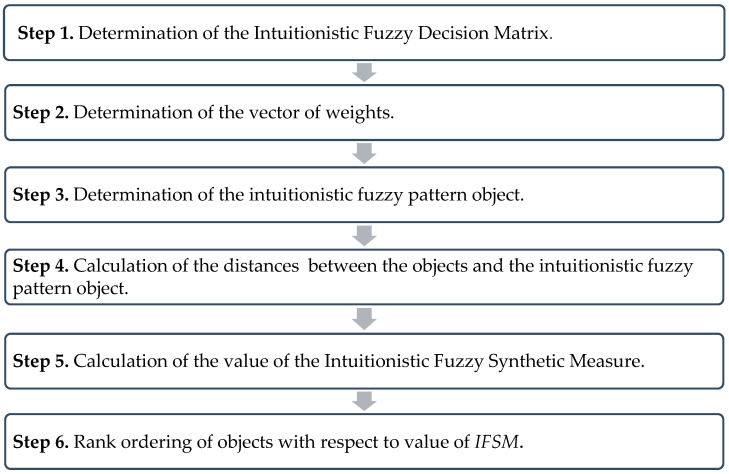
The steps of the IFSM procedure with IF entropy-based weights.

**Figure 2 entropy-25-00961-f002:**
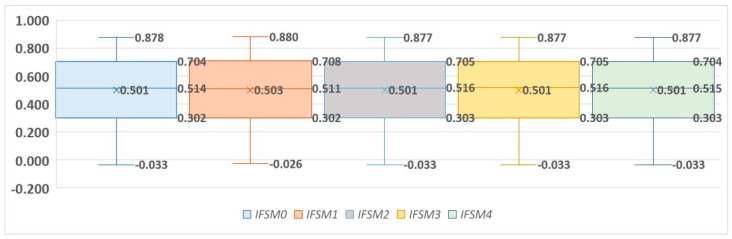
Box plots for the *IFSM* values.

**Table 1 entropy-25-00961-t001:** Assessment of London.

Category *	C_1_	C_2_	C_3_	C_4_	C_5_	C_6_	C_7_	C_8_	C_9_	C_10_
1	2.475%	8.483%	6.754%	4.651%	1.630%	3.856%	7.673%	12.667%	6.632%	9.070%
2	11.307%	18.444%	13.397%	12.601%	5.811%	9.753%	9.008%	31.283%	17.195%	24.155%
3	41.520%	40.197%	42.621%	33.230%	36.469%	54.561%	34.770%	39.145%	48.834%	49.034%
4	42.453%	29.712%	23.608%	45.340%	55.660%	28.366%	26.621%	14.478%	25.731%	16.860%
99	2.244%	3.163%	13.620%	4.179%	0.430%	3.464%	21.928%	2.426%	1.608%	0.882%

* 1—Very unsatisfied, 2—Rather unsatisfied, 3—Rather satisfied, 4—Very satisfied, 99—Don’t know/No Answer/Refuses https://ec.europa.eu/regional_policy/en/information/maps/quality_of_life/ (accessed on 1 December 2022).

**Table 2 entropy-25-00961-t002:** Assessment of London with the use of IFVs.

Parameter	C_1_	C_2_	C_3_	C_4_	C_5_	C_6_	C_7_	C_8_	C_9_	C_10_
Degree of non-membership (ν)	0.138	0.269	0.202	0.173	0.074	0.136	0.167	0.440	0.238	0.332
Degree of membership (μ)	0.840	0.699	0.662	0.786	0.921	0.829	0.614	0.536	0.746	0.659
Degree of hesitancy π	0.022	0.032	0.136	0.042	0.004	0.035	0.219	0.024	0.016	0.009

**Table 3 entropy-25-00961-t003:** IF entropy-based weights obtained using different formulas.

Weight Methods	C_1_	C_2_	C_3_	C_4_	C_5_	C_6_	C_7_	C_8_	C_9_	C_10_
*E*0 (equal weights)	0.100	0.100	0.100	0.100	0.100	0.100	0.100	0.100	0.100	0.100
*E*1 (entropy measure Formula (5))	0.095	0.081	0.105	0.108	0.082	0.098	0.107	0.108	0.109	0.107
*E*2 (entropy measure Formula (9))	0.101	0.100	0.098	0.102	0.098	0.104	0.099	0.097	0.099	0.101
*E*3 (entropy measure Formula (7))	0.101	0.100	0.098	0.102	0.099	0.104	0.099	0.097	0.099	0.101
*E*4 (entropy measure Formula (8))	0.100	0.100	0.099	0.101	0.099	0.102	0.099	0.099	0.099	0.100

**Table 4 entropy-25-00961-t004:** The Garuti’s indexes.

Garuti’s Index	*E*0	*E*1	*E*2	*E*3	*E*4
*E*0	1	0.917	0.904	0.904	0.904
*E*1		1	0.910	0.910	0.908
*E*2			1	0.999	0.990
*E*3				1	0.992
*E*4					1

**Table 5 entropy-25-00961-t005:** The intuitionistic fuzzy pattern object ($I_{IFI}$).

Parameter	C_1_	C_2_	C_3_	C_4_	C_5_	C_6_	C_7_	C_8_	C_9_	C_10_
ν	0.046	0.075	0.088	0.062	0.091	0.036	0.113	0.139	0.063	0.097
μ	0.950	0.802	0.890	0.929	0.816	0.944	0.878	0.856	0.937	0.876
π	0.004	0.123	0.022	0.009	0.093	0.020	0.009	0.005	0.000	0.027

**Table 6 entropy-25-00961-t006:** Values of *IFSMi* coefficients and rankings of cities.

City	IFSM0 Coefficient		IFSM1 Coefficient		IFSM2 Coefficient		IFSM3 Coefficient		IFSM4 Coefficient	
	Value	Rank	Value	Rank	Value	Rank	Value	Rank	Value	Rank
Amsterdam	0.738	7	0.736	7	0.739	7	0.739	7	0.739	7
Ankara	0.449	21	0.461	20	0.448	21	0.448	21	0.448	21
Athina	0.080	35	0.077	35	0.080	35	0.080	35	0.080	35
Belgrade	0.286	28	0.285	28	0.286	28	0.286	28	0.286	28
Berlin	0.615	16	0.618	16	0.617	16	0.616	16	0.616	16
Bratislava	0.354	26	0.354	26	0.355	26	0.355	26	0.355	26
Brussels	0.526	18	0.523	18	0.527	18	0.527	18	0.527	18
Bucharest	0.184	31	0.185	31	0.186	31	0.186	31	0.185	31
Budapest	0.404	24	0.403	24	0.406	24	0.406	24	0.405	24
Dublin	0.696	10	0.695	10	0.696	10	0.696	10	0.696	10
Helsinki	0.856	3	0.856	3	0.856	3	0.856	3	0.856	3
København	0.725	8	0.730	8	0.726	8	0.726	8	0.725	8
Lefkosia	0.431	23	0.437	23	0.429	23	0.429	23	0.430	23
Lisbon	0.439	22	0.438	22	0.440	22	0.440	22	0.439	22
Ljubljana	0.769	5	0.770	6	0.769	5	0.769	5	0.769	5
London	0.673	11	0.674	11	0.674	11	0.674	11	0.673	11
Luxembourg	0.878	1	0.880	1	0.877	1	0.877	1	0.877	1
Madrid	0.353	27	0.353	27	0.354	27	0.354	27	0.353	27
Oslo	0.800	4	0.802	4	0.801	4	0.801	4	0.801	4
Palermo	−0.033	36	−0.026	36	−0.033	36	−0.033	36	−0.033	36
Paris	0.462	20	0.456	21	0.464	20	0.464	20	0.463	20
Podgorica	0.382	25	0.390	25	0.380	25	0.380	25	0.381	25
Prague	0.663	12	0.656	13	0.665	12	0.665	12	0.664	12
Reykjavík	0.649	14	0.654	14	0.648	14	0.648	14	0.649	14
Riga	0.604	17	0.611	17	0.605	17	0.605	17	0.604	17
Rome	0.118	33	0.118	33	0.118	33	0.118	33	0.118	33
Skopje	0.115	34	0.106	34	0.116	34	0.116	34	0.116	34
Sofia	0.256	29	0.256	29	0.258	29	0.258	29	0.257	29
Stockholm	0.766	6	0.774	5	0.767	6	0.767	6	0.767	6
Tallinn	0.707	9	0.712	9	0.708	9	0.708	9	0.707	9
Tirana	0.170	32	0.177	32	0.168	32	0.168	32	0.169	32
Valletta	0.236	30	0.238	30	0.233	30	0.233	30	0.234	30
Vilnius	0.637	15	0.651	15	0.638	15	0.638	15	0.638	15
Warsaw	0.503	19	0.499	19	0.505	19	0.505	19	0.504	19
Wienna	0.871	2	0.877	2	0.872	2	0.871	2	0.871	2
Zagreb	0.662	13	0.668	12	0.663	13	0.663	13	0.663	13
Mean	0.501		0.503		0.501		0.501		0.501	
SD	0.250		0.251		0.251		0.251		0.250	

**Table 7 entropy-25-00961-t007:** Spearman coefficient for *IFSM* measures.

Spearman Coefficient	IFSM0	IFSM1	IFSM2	IFSM3	IFSM4
*IFSM*0	1.0000	0.9992	1.0000	1.0000	1.0000
*IFSM*1		1.0000	0.9992	0.9992	0.9992
*IFSM*2			1.0000	1.0000	1.0000
*IFSM*3				1.0000	1.0000
*IFSM*4					1.0000

**Table 8 entropy-25-00961-t008:** Pearson coefficient for IFSM measures.

Pearson Coefficient	IFSM0	IFSM1	IFSM2	IFSM3	IFSM4
*IFSM*0	1.0000000	0.9997999	0.9999858	0.9999857	0.9999965
*IFSM*1		1.0000000	0.9997361	0.9997301	0.9997685
*IFSM*2			1.0000000	0.9999999	0.9999960
*IFSM*3				1.0000000	0.9999960
*IFSM*4					1.0000000

## Data Availability

Not applicable (for secondary data analysis, see [91]).

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
