# Peer review of "The Impact of the Intuitionistic Fuzzy Entropy-Based Weights on the Results of Subjective Quality of Life Measurement Using Intuitionistic Fuzzy Synthetic Measure"

_entropy, 2023, doi:10.3390/e25070961_

Round 1

Reviewer 1 Report

This paper proposes an extended Intuitionistic Fuzzy Synthetic Measure (IFSM) with intuitionistic fuzzy (IF) entropy-based weights.

The work of this paper is logical.

However, there are some problems to be further improved as well:

1. Authors should carefully check the paper, and correct all the typos and English language errors. This will make the paper more readable and informative.

2. In conclusion, some relevant topics for future work would be helpful for readers.

3. From the data source given in Line 376, the problem of subjective quality of life measurement is not a difficult problem. In my opinion, lots of algorithms can solve this problem. So the necessary of intuitionistic fuzzy (IF) entropy-based weights should be explained.

4. The data in the table can be placed in the appendix.

5. The determination of the intuitionistic fuzzy decision matrix should be explained.

6. A large number of unrelated references are used.

 Thus, I suggest that this paper be accepted after a revision.

Authors should carefully check the paper, and correct all the typos and English language errors. This will make the paper more readable and informative.

Author Response

Answer to the Review

The Authors of the article are most grateful for all the valuable remarks and comments. The suggested changes certainly improved the quality of the article making it more readable and understandable.

Suggestions for Authors 

This paper proposes an extended Intuitionistic Fuzzy Synthetic Measure (IFSM) with intuitionistic fuzzy (IF) entropy-based weights. 

The work of this paper is logical. 

However, there are some problems to be further improved as well: 

  1. Authors should carefully check the paper, and correct all the typos and English language errors. This will make the paper more readable and informative.

The authors check the paper and correct the typos and English language errors.  

  1. In conclusion, some relevant topics for future work would be helpful for readers.

In conclusion, some direction of future research has been added.    

  1. From the data source given in Line 376, the problem of subjective quality of life measurement is not a difficult problem. In my opinion, lots of algorithms can solve this problem. So the necessary of intuitionistic fuzzy (IF) entropy-based weights should be explained.

We agree with the reviewer that lots of methods can be used in solving the problem of quality of life. We mentioned it in the Introduction, but this article is a continuation of research in the field of IFSM application in measuring the subjective quality of life. In previous studies, the same weights were used for the subjective quality of life criteria. In this paper, the assessment of the impact of weighting systems on the stability of IFSM results is considered.

  1. The data in the table can be placed in the appendix.

Tables 3-5 have been moved to the Appendix.

  1. The determination of the intuitionistic fuzzy decision matrix should be explained.

The calculation values in the table have been explained in the example.  The method of determining the fuzzy intuitionistic matrix is described in the article [35]. In the peer-reviewed article, a relevant reference to this work is included in lines 321-322.

  1. A large number of unrelated references are used.

We have checked the article in terms of the literature references paying attention to the consistency of the paper. The references are blocked in the series of different issues under consideration in the manuscript  (quality of life, measurement quality of life,   the intuitionistic fuzzy multi-criteria methods based on reference points, IFSM algorithm, entropy measure, and intuitionistic entropy based-weights).  From the viewpoint of research goals and methodology, it seems consistent. However, they cover several research areas. We want to stick to our method of referencing because it fits in with the concept and layout of our article.

 Thus, I suggest that this paper be accepted after a revision. 

All new fragments in the text are marked in yellow.

Best regards,

Authors

Reviewer 2 Report

The manuscript entitled “intuitionistic fuzzy entropy-based weights” deals with a very important topic in fuzzy number measurement. The employed methods are very promising and the results look very valuable because there is also a comparison against other 4 different IF entropy-based approaches models.

However, the paper could not provide enough novelty and convincible results to show its superiority compared with the existing IF Synthetic Measures mainly due to as follows:

1.  In the introduction section, the existing research gaps did not properly discuss and listed.

2.    The proposed method should be compared with the state-of-the-art IF Synthetic Measures models.

It may be better if the following information is added to the context.

1. Add one Table to show all algorithm's parameters that tuned in this research.

2. There is no flowchart. The authors may add a detailed flow chart to show all steps. 

Author Response

Answer to the Review No. 2:

The Authors of the article are most grateful for all the valuable remarks and comments. The suggested changes certainly improved the quality of the article making it more readable and understandable. 

Comments and Suggestions for Authors  

The manuscript entitled “intuitionistic fuzzy entropy-based weights” deals with a very important topic in fuzzy number measurement. The employed methods are very promising and the results look very valuable because there is also a comparison against other 4 different IF entropy-based approaches models. 

However, the paper could not provide enough novelty and convincible results to show its superiority compared with the existing IF Synthetic Measures mainly due to as follows: 

  1. In the introduction section, the existing research gaps did not properly discuss and listed.

The research gaps are listed in the Introduction.

  1. The proposed method should be compared with the state-of-the-art IF Synthetic Measures models.

The main aim of the article was to investigate the impact of entropy-based criteria weighting methods on IFSM results. The reviewer's suggested comparison of IFSM with other methods for intuitionistic fuzzy sets is also interesting and is listed in the Conclusions as a future study.

We add some information about the most known Intuitionistic fuzzy multiply criteria  techniques based on reference points, such as TOPSIS and VIKOR

It may be better if the following information is added to the context. 

  1. Add one Table to show all algorithm's parameters that tuned in this research.

We add the table with symbols and notation used in the paper.

  1. There is no flowchart. The authors may add a detailed flow chart to show all steps.

The flowchart has been added in Section 3.

All new fragments in the text are marked in yellow.

Best regards,

Authors

Round 2

Reviewer 1 Report

No further comment.

No further comment.

Reviewer 2 Report

The authors have made an important work to clarify and answer to the comments made on the first round of reviewing. The purpose of the paper appears more clearly and this simplifies the overall understanding.